# MetaOCDN: A Cognition-Inspired Meta Optimized Complementary Dual Networks for Online Continual Concept Drift Adaptation

## Abstract

The *Complementary Learning Systems* (CLS) theory points that humans can continuously and efficiently adapt to new tasks through the collaboration between the hippocampus and the neocortex: the former rapidly encodes new knowledge, while the latter extracts structured knowledge by abstract learning. Their synergy enables humans not only to quickly learn new tasks in the short term but also to transfer acquired knowledge across different tasks. Inspired by this theory, we address the challenge of streaming data mining under open environment with concept drift by proposing a cognition-inspired meta optimized complementary dual networks architecture (MetaOCDN), which consists of the Adaptive Fine Tuning Network (AFT-Net) and the Meta Representation Network (MRN-Net). AFT-Net is similar to the hippocampus, selectively fine-tunes key layers based on gradient variations to achieve rapid adaptation to novel concepts; MRN-Net is similar to the neocortex, we design self-supervised duality loss to continuously enhance its deep representation capability, thereby improving generalization to unknown distributions; furthermore, we design MAML-based multi-scale knowledge distillation strategy to facilitate dynamic information flow and knowledge transfer between the two networks. In summary, MetaOCDN provides a brain-inspired collaborative architecture that integrates the rapid responsiveness of AFT-Net with the abstract generalization capacity of MRN-Net, and enhances their interaction through knowledge distillation, thereby achieving a dynamic balance between fast adaptation and stable generalization in non-stationary data streams with concept drift. Extensive experiments demonstrate that MetaOCDN consistently outperforms state-of-the-art baselines across various drift scenarios.

## 1 Introduction

In open environment streaming data mining tasks, concept drift limits model performance. Models trained with traditional batch learning paradigms struggle to quickly adapt to new distribution after concept drift (Lu et al., 2019). At present, researchers expect to train models through online learning approach (Cano & Krawczyk, 2022) (such as active drift detection online learning and adaptive online learning) to capture the dynamic changes in streaming data. The former actively monitors data distribution changes (e.g., via statistical tests or sliding-window error rates) to detect concept drift and performs the targeted update, such as ROALE-DI (Zhang et al., 2020). However, during the process of actively detecting concept drift, the setting of the threshold can significantly affect model performance (Gama et al., 2004). Although the Delayed Detection Index (Liu et al., 2022) alleviates this issue, the false positives, false negatives, and delayed detection remain challenging. The latter overcomes these challenges by adapting models in real time without relying on drift detection, e.g., DDG-DA (Li et al., 2022). However, most of these methods rely on supervised or semi-supervised training strategies, models are difficult to efficiently learn robust features from the limited samples available after concept drift (Liu et al., 2021). They also tend to optimize a single objective, restricting the balance between fast adaptation and generalization.

**How to design model that can quickly adapt after concept drift while having a strong generalization ability to cope with the impact of changes in data distribution?** The *Complementary Learning Systems* (CLS) theory (McClelland et al., 1995; Kumaran et al., 2016) offers new inspi-

ration for us. Humans can quickly extract patterns and adapt to new environments from a limited number of samples, primarily due to the unique structure of the brain: specifically, the neocortex and hippocampus. The CLS theory suggests that the neocortex and hippocampus collaborate to enable efficient learning: the neocortex gradually acquires structured knowledge by alternating between different tasks, and the hippocampus is better at encoding new information quickly. When facing new and complex tasks, the hippocampus retrieves structured knowledge stored in the neocortex to promote rapid learning, and the neocortex encodes the new knowledge from the hippocampus into structured knowledge, it enhances the stability of knowledge and improves the ability to learn quickly. Recent studies have introduced the CLS theory into continual learning and have shown its potential to mitigate catastrophic forgetting (Pham et al., 2023). However, how to transfer this mechanism into open environments for concept drift adaptation remains an open challenge for further exploration. Therefore, to alleviate limitations in existing works, we propose a meta optimized complementary dual network strategy (MetaOCDN). The connection between MetaOCDN and CLS theory is shown in Fig. 1:

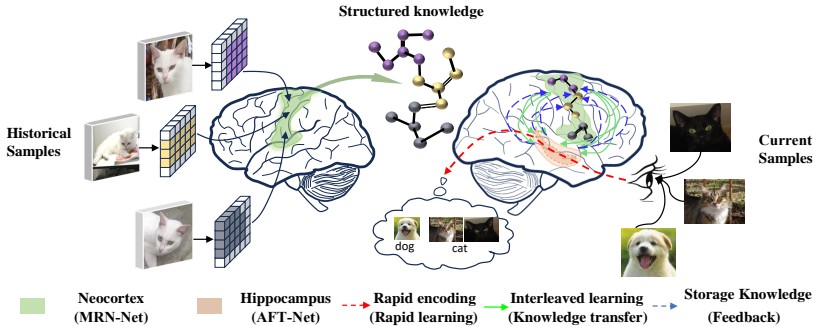

Figure 1: Meta optimized complementary dual network strategy inspired by the CLS theory.

Specifically, we construct Adaptive Fine Tuning Network (AFT-Net) to simulate the hippocampus and design gradient-aware selective fine-tuning strategy to selectively fine-tune its key layers, thereby forming a sparse network. AFT-Net learns task-specific knowledge from the current samples in an online learning manner, to ensure the model rapidly adapts to new distribution. And we construct Meta Representation Network (MRN-Net) to simulate the neocortex, the self-supervised duality loss is designed to continuously refine its feature extraction ability, and offline learning is employed to acquire more robust representations from historical samples. Finally, we design MAML-based multi-scale knowledge distillation strategy to facilitate knowledge transfer from the MRN-Net to the AFT-Net. In conclusion, MetaOCDN achieves rapid adaptation to new distribution while maintaining strong generalization capability. The main contributions of this paper are:

1. Inspired by CLS theory, we propose the MetaOCDN, it includes the AFT-Net and MRN-Net to emulate the hippocampus for rapid learning new knowledge and the neocortex for extracting structured knowledge. The MAML-based multi-scale knowledge distillation strategy further enhances knowledge transfer, balances fast convergence with stable generalization.

2. We analysis why selective fine-tuning the critical layer in the face of different distribution changes has a better effect than fully fine-tuning the network, and at the same time we prove that the MetaOCDN has an excellent sublinear regret bound.

3. The actual performance of MetaOCDN was verified in classification and regression tasks involving concept drift. Compared with the baseline methods, MetaOCDN achieves good results in terms of model convergence speed and generalization after concept drift.

## 2 RELATED WORK

**Active drift detection online learning.** This type of approach mainly relies on dynamic monitoring of model performance or data distribution to determine whether drift has occurred. Typical methods include: Type-LDD (Yu et al., 2023), a pre-trained framework for drift localization and type identification using knowledge distillation; and Targeted EL (Guo et al., 2024), which identifies drift types

and selects base classifiers accordingly to improve diversity, among others. Most of these methods are error-rate–based, relying on window mechanisms and manually set parameters, which often lead to unstable performance (Bifet & Gavalda, 2007). Compared with error rate–based detectors, these methods identify drift timing and location more accurately by comparing data distributions or representation spaces (Liu et al., 2022). Representative approaches include MCDDD (Wan et al., 2024) (contrastive concept embedding), PERCESS (Cai et al., 2025) (latent representation estimation for online prediction), and AMSL (Zhang et al., 2022) (self-supervised adaptive memory). They offer finer-grained detection but rely heavily on representation quality, making them prone to false alarms or delays in real-time streaming scenarios.

**Adaptive online learning.** Adaptive online learning under concept drift bypasses explicit drift detection by assuming that data distribution may change at any time and adapting models through real-time updates. Representative methods include: HBP (Sahoo et al., 2017), which dynamically re-weights network layers to adjust depth during training; OneNet (Wen et al., 2023), which integrates reinforcement learning into online convex optimization to enhance robustness but with limited fast adaptation; ReCDA (Yang et al., 2024), which introduces drift-aware perturbation and representation alignment to learn more stable features; and memory-aware approaches that update parameter importance for continual adaptation (Aljundi et al., 2018). Overall, these methods improve adaptability and robustness under drift through dynamic adjustment, yet most rely on supervised or semi-supervised training and struggle to efficiently learn from limited post-drift samples, with objectives often biased toward either fast adaptation or generalization, but not both.

## 3 METAOCDN: COGNITION-INSPIRED ONLINE LEARNING ALGORITHM

Concept drift is a phenomenon in which the statistical properties of a target domain change over time in an arbitrary way (Lu et al., 2019). Given a time period $[0, t]$, there is a set of streaming data $DS = (X_t, y_t)$, $X_t$ denote the feature vector at the timestamp $t$, $y_t$ denote the corresponding label. The streaming data follow a certain distribution $F_{0,t}(X, y)$, concept drift occurs at timestamp $t + 1$, if $F_{0,t}(X, y) \neq F_{t+1,+\infty}(X, y)$, denoted as $\exists t : P_t(X, y) \neq P_{t+1}(X, y)$. In addition, we denote the current samples as $\mathcal{D}^t = (x_i^t, y_i^t)$, the historical samples as $\mathcal{D}^m = (x_i^m, y_i^m)$, and $\{i = 1, 2, \ldots, n\}$.

### 3.1 ADAPTIVE FINE TUNING NETWORK

According to the CLS theory, the hippocampus's rapid learning ability primarily stems from two aspects: (1) its synapses exhibit strong plasticity, can quick adjust after one or a few learning trials; and (2) it encodes new information through sparse neuronal activation patterns. To simulate this mechanism, we enhance the plasticity of AFT-Net via online learning and design a gradient-aware selective fine-tuning strategy to construct a sparse network.

Similar to the hippocampus, online learning incrementally learns from streaming data, updates parameters in real time and adapts to current samples distribution within a few iterations. Accordingly, the AFT-Net is trained under the online learning paradigm (Bartlett et al., 2007), with its parameters are updated via online gradient descent: $\theta_{t+1} = \theta_t - \eta \nabla_\theta \mathcal{L}^{AFT}(\theta_t; \mathcal{D}^t)$, $\eta$ denotes the learning rate, and $\mathcal{L}^{AFT}$ represents the total loss of AFT-Net. Relying solely on online learning to enhance rapid adaptation to new distribution is insufficient. As indicated by online gradient descent, processing each current sample requires updating all parameters, resulting in a computational complexity of $O(d)$. This not only increases the computational burden but also leads to overfitting to new distribution and forgetting of previously learned knowledge.

To better simulate the hippocampus and accelerate model convergence, we conduct lots of experiments on three standard concept drift datasets. As a tool for loss minimization, gradients can more intuitively and precisely reveal the model's sensitivity to changes in data distribution. The results show that gradients provide a more accurate characterization of the model's state after concept drift—different types and degrees of drift exert significantly different impacts on various layers of the model (see Fig. 2). So we design a gradient-aware selective fine-tuning strategy that freezes parameters insensitive to the new distribution, thereby constructing a sparse AFT-Net.

Firstly, when the AFT-Net is trained at timestamp $t$, the gradient of the $l$-th layer is denoted as $g_t^l$, in this paper, we use the gradient norm $\left\|g_t^l\right\|_2$ to represent the changes of the $l$-th layer. To capture the long-term gradient variation patterns of the model, we design a historical gradient variation rate

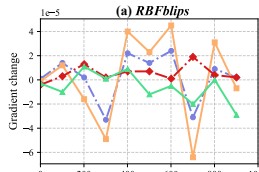 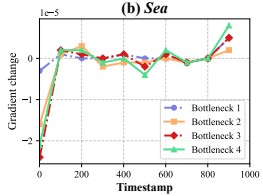 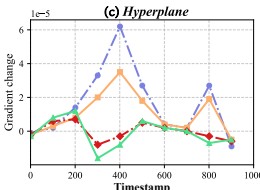

Figure 2: *Gradient changes of network layers. We analyze gradient changes of ResNet on datasets with different drift types: abrupt (RBFBlips), gradual (Sea), and incremental (Hyperplane), with drifts occurring at timesteps 250, 500, and 750.*

matrix $\mathbf{G} \in \mathbb{R}^{m \times L}$ to store the model's historical gradient variation rates of all $L$ layers over the last $m$ timestamps, its element $r_t^L = \left\| g_t^L \right\|_2 - \left\| g_{t-1}^L \right\|_2$ is the rate of change of the gradient. On this basis, we design layer gradient sensitivity index $\mathcal{R}_t^l$ to reveal the influence intensity of different layers:

$$\mathcal{R}_t^l = \frac{\left\| g_t^l \right\| \cdot f\left( r_t^l, \sigma^l \right)}{\sum_{i=1}^{L} \left\| g_t^i \right\| \cdot f\left( r_t^i, \sigma^i \right)} \tag{1}$$

Among them, $\sigma^l$ is the standard deviation of historical gradient variation rate and it is used to automatically "balance" the contribution of each layer to the overall measurement. Adaptively adjust the weights $f(r_t^l, \sigma^l) = \exp(r_t^l/\sigma^l)$, a larger value of $r_t^l/\sigma^l$ indicates that the $l$-th layer is more sensitive to changes in the new distribution, conversely, the more stable it is.

Finally, a drift-aware threshold is dynamically generated for each layer to determine whether the layer should be frozen: $\tau_t^l = \bar{\mathcal{R}}_t^L + \sigma_t^2$, and $\bar{\mathcal{R}}_t^L = 1/L \cdot \sum_{l=1}^{L} \mathcal{R}_t^l$. When $\mathcal{R}_t^l < \tau_t^l$, the $l$-th layer is well-adapted to current samples, thus is frozen to avoid unnecessary resource consumption; otherwise, the layer is regarded as more sensitive to the new distribution and is activated for local updates. By retaining only the layers sensitive to distribution changes, the model forms a sparse network. When concept drift occurs, only these key layers need to be fine-tuned, thereby improving response efficiency while effectively mitigating overfitting.

## 3.2 META REPRESENTATION NETWORK

Similarly, MetaOCDN constructs a Meta Representation Network (MRN-Net) that learns structured knowledge from historical samples, analogous to the neocortex. Neocortex relies on slow and cumulative synaptic adjustments, allowing it to extract stable patterns through long-term, cross-task learning and form task-agnostic structured knowledge. Inspired by this, we design a self-supervised duality loss (Silva et al., 2024) to optimize the model's representation ability, thereby building MRN-Net capable of "learn to learn extract features".

Firstly, we use the Wasserstein distance to measure the similarity between current and historical samples in order to select appropriate training instances (Chizat et al., 2020). Based on this we divide them into positive samples $\mathcal{D}^{m+}$ and negative samples $\mathcal{D}^{m-}$. We design self-supervised duality loss to optimize the representation capability of the MRN-Net. The self-supervised duality loss does not rely on samples' labels, which is crucial for label-scarce streaming data. It helps the model learn more discriminative and robust feature representations, enables the MRN-Net to better capture the underlying structure of the data.

Specifically, the self-supervised duality loss consists of similarity loss and difference loss. We leverage MRN-Net to jointly represent the positive samples $z^+$ and the current samples $z^t$, and approximate the similarity loss by maximizing the mutual information $\max\limits_{\varphi} I(z^+; z^t)$ between them. We approximate the maximization of mutual information by maximizing its lower bound (Oord et al., 2018), denoted as: $I_{\text{Low}}(z^+; z^t) = \mathbb{E}_{p(z^+, z^t)} \log \frac{p(z^+|z^t)}{p(z^+)} \geq \mathbb{E}_{p(z^+, z^t)} \log \frac{q(z^+|z^t)}{p(z^+)}$. Since computing the lower bound of mutual information is challenging, we adopt InfoNCE as a surrogate objective for mutual information maximization. We have:

$$\ell^{\text{sim}} = -I_{\text{Low}}\left( z^+; z^t \right) = -\frac{1}{n} \sum_{j=1}^{n} \log \frac{e^{\psi\left(z_j^t, z_j^+\right)}}{\sum_{i=1}^{n} e^{\psi\left(z_j^t, z_i^+\right)} + \xi} \tag{2}$$

$\psi\left(\cdot\right)$ denotes the similarity function, $n$ is the number of samples, and $\xi$ is a stability term that smooths the loss function. The proof is in Appendix A.1.

To further enhance MRN-Net's ability to discriminate irrelevant features, we construct the difference loss by minimizing the mutual information between negative samples and the current samples representations $\min_{\varphi} I(z^-; z^t)$. Similarly, we use an upper bound on mutual information as an approximation for this minimization (Zhang et al., 2023). By introducing a random variable $\mathcal{N}$ ($\mathcal{N}$ is sampled from the original input of negative samples), we can derive the upper bound of mutual information, which is expressed as: $I\left(z^-; z^t\right) = I\left(z^-; z^t; \mathcal{N}\right) + I\left(z^-; z^t \mid \mathcal{N}\right)$. From the derivation in Appendix A.1, the difference loss is given by:

$$\ell^{diff} \approx D_{KL}\left(p\left(z^- \mid \mathcal{N}\right) \| q\left(z^-\right)\right) + D_{KL}\left(p\left(z^t \mid \mathcal{N}\right) \| q\left(z^t\right)\right) \tag{3}$$

In conclusion, the total loss of the MRN-Net is: $\mathcal{L}^{\text{MRN}} = \beta\ell^{sim} + (1-\beta)\ell^{diff}$. $\beta$ is a hyperparameter that balances the two losses.

### 3.3 MAML-based Multi-scale Knowledge Distillation

Finally, the CLS theory suggests that the human brain integrates rapid learning and resistance to forgetting through the synergy between the hippocampus and the neocortex: the hippocampus rapidly encodes information and replays it during sleep, while the neocortex repeatedly extracts structured knowledge and feeds it back to the hippocampus to accelerate learning. Inspired by this, we design MAML-based multi-scale knowledge distillation strategy (Finn et al., 2017): AFT-Net adapts via inner-loop updates with replayed historical samples and transfers knowledge to the MRN-Net, which extracts cross-task stable patterns and feeds them back, completing the outer loop. This "replay–extract–transfer–feedback" synergy enables MetaOCDN to achieve both fast adaptation and long-term generalization in dynamic environment.

Specifically, we divide the feature maps extracted by the AFT-Net and the MRN-Net (denoted as $F^{\text{AFT}}, F^{\text{MRN}} \in \mathbb{R}^{H \times W \times C}$) into multi-scale units, and aggregate the knowledge within each unit through average pooling:

$$\Pi_{p_i}^{ATF} = \frac{1}{p_i^2} \sum_{h,w \in (H,W)} F^{ATF}(h,w), \qquad \Pi_{p_i}^{MRN} = \frac{1}{p_i^2} \sum_{h,w \in (H,W)} F^{MRN}(h,w) \tag{4}$$

$p_i \in \{p_1, p_2, \ldots, p_K\}$ represents a set of different scales, $\prod_{p_i}^{MRN} \in \mathbb{R}^{p_i \times p_i \times C}$ represents the aggregated features at different scales. Then, we concatenate the aggregated features from different scales along the channel dimension to form the final multi-scale knowledge representation:

$$\Pi_{\text{fused}}^{AFT} = \text{Concat}\left(\Pi_{p_1}^{AFT}, \ldots, \Pi_{p_K}^{AFT}\right), \qquad \Pi_{\text{fused}}^{MRN} = \text{Concat}\left(\Pi_{p_1}^{MRN}, \ldots, \Pi_{p_K}^{MRN}\right) \tag{5}$$

Distillation loss is expressed as follows: $\ell^{KD} = \text{KL}\left(\text{softmax}\left(\Pi_{\text{fused}}^{ATF}\right), \text{softmax}\left(\Pi_{\text{fused}}^{MRN}\right)\right)$. The interaction between the neocortex and hippocampus relies not only on knowledge transfer but also on memory replay and structured knowledge extraction. Inspired by this, we introduce MAML to optimize the knowledge distillation process and better simulate their synergistic mechanism.

Specifically, we map the AFT-Net and the MRN-Net into the bi-level optimization framework of MAML. The AFT-Net serves as the inner-loop optimizer, trains on replayed information provided by the MRN-Net; meanwhile, the MRN-Net acts as the outer-loop optimizer, extracts structured knowledge based on the update dynamics of the AFT-Net and feeding it back. Through this dual-loop process, the MRN-Net can perceive and adapt to the learning state of the AFT-Net, distilling more tailored knowledge to enhance its adaptability.

The initialization parameters of AFT-Net are $\theta$. For the $i$-th inner-loop optimization, the parameter update of AFT-Net is denoted as $\theta^i$. Specifically, support sets $\mathcal{D}^s$ are randomly sampled from historical samples, and AFT-Net is iteratively updated via stochastic gradient descent. For example, with a single gradient update: $\theta^i = \theta - \alpha_{in} \frac{\partial \ell^{KD}(\mathcal{D}^s; \theta, \varphi)}{\partial \theta}$, $\alpha_{in}$ denotes the learning rate of the AFT-Net. After multiple rounds of information replay, the MRN-Net serves as the outer-loop optimizer to extract structured knowledge. We employ a regularization term as an approximate gradient to transfer the knowledge encoded in the AFT-Net parameters to the MRN-Net, as follows: $\varphi =$

$\varphi - \frac{\alpha_{out}}{T^{out}} \sum_{i \in T^{out}} \left|\left| \varphi - \theta^i \right|\right|^2$, where $\alpha_{out}$ denotes the learning rate of the MRN-Net, and $T^{out}$ represents the training epoch. Finally, the knowledge of the MRN-Net is fedback to the AFT-Net:

$$\theta_{t+1} = \theta_t - \lambda_\theta \nabla_\theta \left( \sum \ell^{cross}(\mathcal{D}^t, f(\theta_t)) + \ell^{KD}(\mathcal{D}^t; \theta_t, \varphi_t) + R(\varphi_t, \theta_t) \right) \tag{6}$$

Here, $\ell^{cross}(\cdot)$ denotes the loss of the model on the current samples after multiple rounds of information replay, and $R(\varphi_t, \theta_t)$ represents the regularization term. Since the parameters of the MRN-Net contain a large amount of meta knowledge and exhibit strong adaptability to changes in data distribution, we align the parameter spaces of the two networks and introduce a regularization penalty to constrain the boundaries of the AFT-Net's parameters. By incorporating this parameter alignment mechanism, the model complexity is reduced while effectively mitigating instability during online training, thereby enhancing the model's ability to rapidly adapt to distribution changes.

## 4 MODEL PERFORMANCE ANALYSIS

To better understand how the gradient-aware selective fine-tuning strategy can accelerate the adaptation speed of MetaOCDN, we conduct a theoretical analysis of it. At the same time, we prove the efficiency of MetaOCDN through its regret bound.

### 4.1 ANALYSIS OF GRADIENT-AWARE SELECTIVE FINE TUNING

For MetaOCDN, there are two main update strategies: (1) selectively adjusting the key layers with significant gradient fluctuations, and (2) full fine-tuning all model parameters. However, full fine-tuning not only tends to cause overfitting on the limited number of target samples and catastrophic forgetting, but also hinders knowledge transfer, while reducing the model's ability to rapidly adapt to current samples (Lee et al.), so we analyze it.

The parameters of AFT-Net are denoted as $\theta$. On stationary streaming data (historical samples), the model loss approaches zero, i.e., $\mathcal{L}^{AFT}(\theta_t, \mathcal{D}^m) \to 0$. We set the selective fine-tuning's loss is $\mathcal{L}^{ft}(\theta_t, \mathcal{D}^t)$, for gradient-aware selective fine-tuning, adaptation to current samples is achieved primarily by updating the layers with large fluctuations, and update process is expressed as follows:

$$\partial_t \theta^{sle} = -\nabla_{\theta^{sle}} \mathcal{L}^{ft}\left(\theta^{sle}, \mathcal{D}^t\right), \partial_t \theta^{oth} = 0 \tag{7}$$

Let $\theta^{sle}$ denote the network parameters selected, and $\theta^{oth}$ denote the parameters that remain unchanged. For full fine-tuning, all layer parameters are updated in:

$$\partial_t \theta^{sle} = -\nabla_{\theta^{sle}} \mathcal{L}^{ful}(\theta^{sle}, \mathcal{D}^t), \partial_t \theta^{oth} = -\nabla_{\theta^{oth}} \mathcal{L}^{ful}(\theta^{oth}, \mathcal{D}^t) \tag{8}$$

**Theorem 1.** When facing concept drift of varying degrees and types, for any $\delta > 0$, there exists at least a probability such $1 - \delta$ that the convergence loss of selective fine-tuning the chosen layers is 0, while the loss caused by full fine-tuning is greater than that of selective fine-tuning. In Appendix A.2, we will prove this conclusion.

### 4.2 ANALYSIS OF THE REGRET BOUNDARY

We primarily focus on the performance of the AFT-Net. Let $\theta_1$ and $\theta_2$ denote the parameters of the AFT-Net at two arbitrary timestamps. For notational convenience, we use $f(\theta)$ to represent the loss function $\mathcal{L}^{AFT}$ and impose the following assumptions on it.

**Assumption 1** (*Lipschitz Continuity*): The loss function $f(\theta)$ is Lipschitz continuous with respect to the parameter $\theta$. According to the bounded gradient criterion, $|| \nabla f(\theta) || \le l$.

**Assumption 2** (*Bounded Parameter Domain*): The parameter domain $\mathcal{W}$ has a diameter of $\Gamma$, i.e., for arbitrary AFT-Net and MRN-Net parameters $\varphi$ and $\theta$: $|| \varphi - \theta || \le \Gamma, \forall \varphi, \theta \in \mathcal{W}$.

These assumptions are largely standard in online learning Cesa-Bianchi & Lugosi (2006), and they are particularly applicable to model adaptation problems in dynamic environment. Specifically, **Assumption 1** avoids the optimization instability caused by changes in data distribution, ensuring that the gradient does not explode due to sudden distribution changes when the model is updated, while **Assumption 2** provides a feasible framework for theoretical analysis (such as the upper bound of

Regret). In the context of strong convex functions, these assumptions lead to sublinear convergence rates, so in the Appendix A.3 we prove that the loss function $f(\theta)$ is strong convex.

The regret bound is often used to measure the performance of online learning and is defined as the difference between the cumulative loss of the algorithm in round decision-making and the cumulative loss of the optimal model in the assumption space. Since the AFT-Net uses online gradient descent to update parameters $\theta_{t+1} = \theta_t - \eta \nabla_\theta f(\theta_t)$, we analyzed it's regret bound. The regret boundary of the AFT-Net can be expressed as (Demšar, 2006):

$$regret = \sum_{t=1}^{T} f_t(\theta_t) - \min_{\theta \in \mathcal{W}} \sum_{t=1}^{T} f_t(\theta) = O(\frac{(l_1 + \beta_1 \Gamma)^2}{2\delta} \ln T) \tag{9}$$

$l_1$ is the boundary of the gradient, $\theta_t$ is the AFT-Net parameter at the current moment, $\theta$ represents the optimal model parameters within the hypothesis space, $\min_{\theta \in \mathcal{W}} \sum_{t=1}^{T} f_t(\theta)$ is the cumulative loss in the decision-making of the optimal model round. The proof of Equation 9 is given in the Appendix A.4, we prove that the AFT-Net has a regret bound approximately equal to $O(\ln T / 2\delta)$. It indicates that it can converge to a very good effect within step $T$.

## 5 EXPERIMENTS

**Experiment Setting.** To comprehensively evaluate the MetaOCDN model, we validated its performance on both classification and regression tasks. For the classification task, we used six datasets, comprising standard concept drift benchmarks (*RBFblips*, *Sea*, *Hyperplane*) and real-world datasets (*Kddcup99*, *MIRS*, *Yoga*). For the regression task, we utilized three real-world datasets: *ETTH2*, *Ettm1*, and *WTH*. Detailed information on all datasets and comparison methods are provided in Appendix B.3. Notably, the AFT-Net and MRN-Net models, used for comparison in this paper, are both built upon a ResNet12 backbone. Further experimental settings, such as model parameters, are detailed in Appendix B.1.

### 5.1 COMPARISONS WITH PRIOR WORK

We compared the performance of MetaOCDN and other methods on the classification task and the regression task. For the classification task, the average real-time accuracy (*Avgracc*) and cumulative accuracy (*Fincacc*) were used as evaluation indicators (see Appendix B.4). For the regression task, we used MSE and MAE as evaluation indicators. The results are as shown in Table 1:

Table 1: Comparison of different methods on classification and regression tasks.

| | Classification (*Avgracc*) | | | | | | Regression (*MSE*) | | | AvgRank |
|---|---|---|---|---|---|---|---|---|---|---|
| | **RBFblips** | **Sea** | **Hyperplane** | **Kddcup99** | **MIRS** | **Yoga** | **ETTH2** | **ETtm1** | **WTH** | |
| DWM | 55.40(16) | 69.07(11) | 87.20(3) | 83.60(5) | 44.71(15) | 52.54(4) | 9.596(9) | 7.949(9) | 0.904(4) | 8.44 |
| OBC | 88.05(7) | 60.68(15) | 74.59(13) | 96.41(2) | 48.94(14) | 47.04(15) | 8.478(8) | 5.073(10) | — | 10.5 |
| RUS | 90.58(6) | 61.00(14) | 73.37(14) | 15.98(17) | 61.51(2) | 48.92(12) | 43.69(10) | 67.403(12) | 10.664(11) | 11.11 |
| LEV | 93.27(5) | 60.51(16) | 71.25(16) | 96.03(3) | 58.00(8) | 43.45(17) | 54.548(11) | 25.013(11) | 10.00(10) | 11.22 |
| ARF | 83.27(12) | 67.06(12) | 77.33(11) | 99.38(1) | 59.92(6) | 51.14(7) | 50.9(12) | 22.54(10) | 4.11(8) | 8.78 |
| DNN | 87.16(8) | 71.55(10) | 85.78(6) | 71.86(9) | 50.13(13) | 49.84(11) | 178.8(13) | 91.59(14) | 90.69(15) | 11.33 |
| ResNet | 83.00(13) | 74.48(8) | 86.37(5) | 65.35(10) | 37.75(17) | 46.32(16) | 801.9(14) | 225.1(15) | 47.58(13) | 12.44 |
| Highway | 84.82(9) | 76.84(5) | 88.41(2) | 75.37(8) | 53.48(11) | 51.54(5) | 775.6(16) | 81.94(13) | 2875.1(16) | 9.33 |
| HBP | 93.50(4) | 77.71(3) | 86.92(4) | 76.70(7) | 54.13(10) | 53.60(3) | 685.4(15) | 232.63(17) | 40.56(12) | 8.22 |
| DenseNet | 94.42(2) | 75.44(6) | 89.05(1) | 87.56(4) | 60.87(4) | 54.13(2) | 801.92(17) | 225.11(16) | 47.58(14) | 7.22 |
| Informer | 57.67(15) | 72.43(9) | 76.11(12) | 23.31(11) | 52.64(12) | 48.85(13) | 1.69(7) | 1.18(7) | 1.10(6) | 10.56 |
| ER | 84.15(10) | 76.89(4) | 81.47(10) | 23.01(15) | 60.87(5) | 50.84(8) | 0.264(6) | 0.149(5) | 1.074(5) | 7.44 |
| DER++ | 83.45(11) | 74.48(8) | 71.79(15) | 23.27(12) | 58.72(7) | 50.47(9) | 0.1742(4) | 0.092(3) | 4.156(9) | 8.89 |
| FsNet | 93.99(3) | 78.21(2) | 84.23(7) | 22.56(16) | 61.07(3) | 50.35(10) | 0.069(2) | 0.163(6) | 1.732(7) | 6.44 |
| Time-TCN | 58.63(14) | 61.11(13) | 84.23(7) | 23.24(13) | 57.93(9) | 51.27(6) | 0.234(5) | 0.101(4) | 0.553(3) | 8.44 |
| PatchTST | 26.75(17) | 39.8(17) | 49.8(17) | 23.2(14) | 44.38(16) | 48.52(14) | 0.138(3) | 0.077(2) | 0.224(1) | 11.22 |
| **MetaOCDN** | **97.62(1)** | **79.28(1)** | 82.64(9) | 82.11(6) | **61.92(1)** | **54.24(1)** | **0.039(1)** | **0.031(1)** | **0.27(2)** | **2.55** |

As shown in Table 1, our proposed method performs well on synthetic datasets exhibiting abrupt and gradual concept drift, but performs relatively poorly on the incremental drift dataset *Hyperplane*. This is because incremental drift spans a long duration and changes only slightly over time without clear drift points. As a result, during the model update process, the AFT-Net tends to freeze more layers, preventing timely updates that would allow it to capture subtle distribution shifts, thereby degrading performance. Meanwhile, on real-world datasets, our method achieves good results on *MIRS* and *Yoga*, but performs less effectively on *Kddcup99*. This is primarily because *Kddcup99*

consists of discrete features, while neural networks are black-box models and often struggle to interpret such discrete attributes. In contrast, ARF, based on the recursive splitting mechanism of random forests, can naturally adapt to the partitioning of discrete feature spaces. Its information gain criterion is inherently compatible with categorical variables, enabling it to achieve superior performance on such datasets. In the regression task, MetaOCDN demonstrates strong performance. ResNet enhances the training of deep models through its residual structure, enabling it to capture complex patterns in time series data. Additionally, the MRN-Net extracts rich structural representations from historical samples, providing a significant advantage when modeling time series data.

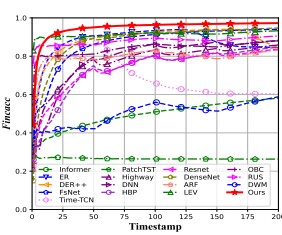 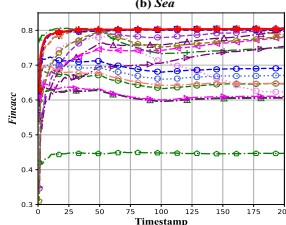 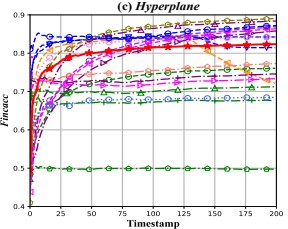

Figure 3: Comparison of *Fincacc* of different methods

Fig. 3 shows the *Fincacc* of each algorithm over different time steps. Similarly, MetaOCDN performs poorly on the *Hyperplane* but achieves good results on the remaining datasets. The remaining experimental results are in Appendix B.5.

**Statistical Analysis.** This paper also employs the Bonferroni-Dunn test to evaluate the statistical significance of differences (Critical Difference) among all methods. According to the calculation, under the significance level $\alpha = 0.05$, the critical difference (CD) is 6.72. The statistical analysis results are shown in Fig. 4. In the figure, methods that do not show a significant difference are connected with red lines. The results indicate that, from a statistical perspective, the method proposed in this chapter demonstrates a clear advantage.

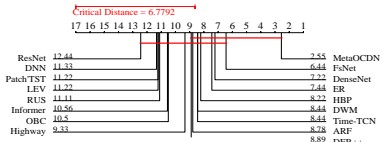

Figure 4: Bonferroni-Dunn test of all methods

To evaluate the convergence speed of MetaOCDN after concept drift occurs, this section compares and analyzes the recovery performance at different drift points. During the determination of convergence points, the convergence threshold is set to $\varepsilon = 0.8$. Table 2 presents the convergence performance of various algorithms on five datasets with known drift points. In the table, each row lists three values representing the recovery scores of each algorithm at the early, middle, and late drift points, respectively. "-" indicates that the model fails to learn features to fit the data at that drift point.

Table 2: *RSA* comparison of different methods

| Datasets | DWM | OBC | RUS | LEV | ARF | DNN | ResNet | Highway | HBP |
|---|---|---|---|---|---|---|---|---|---|
| *RBFblips* | 2.16/0.15/0.12 | 0.54/0.49/0.26 | 0.87/0.56/0.82 | 0.68/0.27/0.31 | -/0.28/0.71 | 0.14/0.11/0.10 | 0.63/0.11/- | 0.10/0.12/- | 0.13/0.10/0.06 |
| *Sea* | 1.10/1.16/0.30 | 1.37/1.15/0.38 | 1.37/1.13/0.35 | 1.4/0.38/1.17 | 0.5/1.0/0.33 | 1.93/1.50/0.31 | 1.78/0.63/0.22 | 1.70/1.00/0.21 | -/2.17/0.30 |

| Datasets | DenseNet | Informer | ER | DER++ | FsNet | Time-TCN | PatchTST | **Ours** |
|---|---|---|---|---|---|---|---|---|
| *RBFblips* | 0.46/0.11/0.25 | 0.51/0.40/1.01 | 0.11/0.17/0.83 | 0.45/0.15/0.03 | 0.05/0.07/0.31 | -/0.77/1.45 | -/0.76/1.45 | **0.13/0.03/0.02** |
| *Sea* | 0.23/0.56/0.30 | 0.54/0.23/0.23 | 0.21/0.64/0.21 | 0.23/0.21/0.19 | 0.22/0.48/0.20 | 0.63/0.61/0.60 | 0.63/0.60/0.59 | **0.21/0.43/0.17** |

Table 2 shows that MetaOCDN converges well on datasets with two known drift points, quickly regaining high accuracy after drift. This benefit stems from the gradient-aware selective fine-tuning strategy, which focuses updates on distribution-sensitive layers and thus achieves faster convergence.

## 5.2 ABLATION EXPERIMENT

**Gradient-aware selective fine-tuning analysis.** Fig. 5 illustrates the gradient variations of the four residual blocks in AFT-Net on benchmark datasets with concept drift. Based on this, we evaluate the convergence speed of AFT-Net under different residual block freezing settings to validate the effectiveness of gradient-aware selective fine-tuning analysis.

We present the results of the model on the *RBFBlips*, with the remaining datasets provided in Appendix B.6. The line plots depict the gradient variations of the four residual blocks around three different drift points; the green bars illustrate the convergence speed when different residual blocks are frozen; and the blue bars compare model performance and parameter updates between selective fine-tuning and full fine-tuning. Experimental results show that freezing residual blocks with large gradient fluctuations diminishes the model's rapid adaptation ability, whereas gradient-aware selective fine-tuning not only achieves higher accuracy than full fine-tuning but also significantly reduces parameter overhead.

We also compared the convergence speed and parameter scale of MetaOCDN's gradient-aware selective fine-tuning strategy with full fine-tuning on real-world datasets.

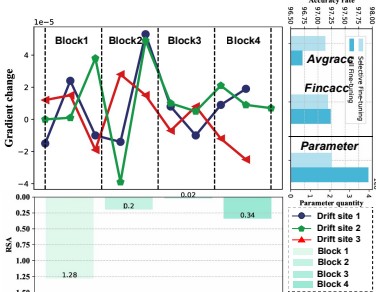

Figure 5: Gradient variation and result analysis

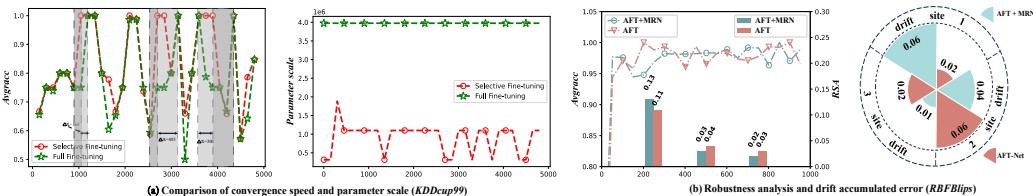

Figure 6: A partial ablation study results figure.

Fig. 6(a) presents a comparison of the model under two update strategies in terms of convergence speed and parameter overhead. The gradient-aware selective fine-tuning strategy enables the model to converge to superior performance within a shorter time while significantly reducing the number of parameters required for updates, thereby improving training efficiency and resource utilization without sacrificing accuracy.

**Robustness Analysis of MRN-Net.** We compared the adaptability of MetaOCDN under MRN-Net and AFT-Net collaboration versus AFT-Net alone on three datasets with explicit drift points. The evaluation metrics include *RSA* (Recovery Speed after Adaptation), which measures the model's real-time convergence ability during drift, and *DCE* (Drift Cumulative Error), which captures the accumulated error during the drift adaptation phase. Partial results are shown in Fig. 6(b), with the remaining results provided in Appendix B.6. Experimental results indicate that MetaOCDN with both networks collaborating exhibits significantly smaller overall accuracy fluctuations. During changes in data distribution, MRN-Net provides more robust initialization or adjustment signals for the online adaptation process, enabling the model to converge more quickly to the new distribution while substantially reducing accumulated error during the drift adaptation phase.

## 6 CONCLUSION

Inspired by the theory of *Complementary Learning Systems*, we propose MetaOCDN. This approach constructs a meta optimized complementary dual network architecture consisting of an Adaptive Fine-Tuning Network (AFT-Net) and a Meta-Representation Network (MRN-Net), analogous to the cooperative mechanism between the hippocampus and neocortex in the human brain. To address the challenge of concept drift in open environments, we focus on enhancing the model's rapid adaptation capability and improving its robustness, which effectively mitigates instability during online training and boosts overall performance under dynamic data distributions.

REPRODUCIBILITY STATEMENT

For reproducibility, we elaborate on the overall pipeline of our work in Section 3. And in Appendix B.1, we provide a description of the model architecture and key parameter settings. In the future, we will upload the source code to a public GitHub repository.

ETHICS STATEMENT

MetaOCDN aims to improve the robustness and adaptability of models in streaming data mining tasks with concept drift, which could be beneficial in the real world, such as financial analysis and anomaly detection as described. All experiments were based on publicly available standard datasets and did not involve any personal privacy or sensitive information. They also did not involve human or animal experiments and did not require additional ethical approval.

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

# A APPENDIX

## A.1 PROOF OF SECTION 3.2

In Section 3.2, we construct a Meta Representation Network (MRN-Net) to learn structured knowledge from historical samples. To enhance the model's representation capability, we design a self-supervised duality loss, which consists of similarity loss and difference loss. The similarity loss reinforces representation consistency among similar samples, while the difference loss pushes apart representations of unrelated samples. This dual mechanism ensures semantic clustering while improving feature discriminability, leading to more robust and generalizable representations.

**Self-supervised similarity loss.** We estimate a lower bound of mutual information to enable the model to capture shared features. The mutual information lower bound is expressed as follows:

$$I_{Low}(z^+; z^t) = \mathbb{E}_{p(z^+, z^t)} \log \frac{p(z^+ \mid z^t)}{p(z^+)} \geq \mathbb{E}_{p(z^+, z^t)} \log \frac{q(z^+ \mid z^t)}{p(z^+)} \tag{10}$$

where $p(z^+ \mid z^t)$ is the conditional distribution of $z^+$ given $z^t$, $\mathbb{E}_{p(z^+, z^t)}$ is the expectation under the joint distribution $p(z^+, z^t)$, and $q(z^+ \mid z^t)$ denotes the variational distribution that $p(z^+, z^t)$. Specifically, we independently sample a set of samples $\{z_1^+, \cdots, z_n^+\}$ from the proposal distribution $\pi(z^+)$, and assign the importance weight $w_{z^+} = \frac{e^{\psi(z^t, z^+)}}{\sum_i^n e^{\psi(z^t, z_i^+)}}$. $\psi(z^T, z_i^+)$ is the cosine similarity. Given the sample set and the target sample, $q(z^+ \mid z^t)$ can be replaced by:

$$q(z^+ \mid z^t, z_{1:n}^+) = \pi(z^+) \cdot \frac{n \cdot e^{\psi(z^t, z^+)}}{e^{\psi(z^t, z^+)} + \sum_{i=2}^n e^{\psi(z^t, z_i^+)}} \tag{11}$$

In summary, the mutual information lower bound is given by (Sordoni et al., 2021):

$$
\begin{aligned}
I_{Low}(z^+; z^t) &\geq \mathbb{E}_{p(z^+, z^t)} \log \frac{q(z^+ \mid z^t, z_{1:n}^+)}{p(z^+)} \\
&\geq \mathbb{E}_{p(z^+, z^t)} \left\lfloor \mathbb{E}_{\pi(z_{1:n}^+)} \log \frac{np(z^+) \cdot w_{z^+}}{p(z^+)} \right\rfloor \\
&= \mathbb{E}_{p(z^+, z^t)} \lfloor \mathbb{E}_{\pi(z_{1:n}^+)} \log \frac{n \cdot e^{\psi(z^t, z^+)}}{e^{\psi(z^t, z_1^+)} + \sum_{i=2}^n e^{\psi(z^t, z_i^+)}} \rfloor \\
&= \mathbb{E}_{p(z^+, z^t)\pi(z_{1:n}^+)} \left\lfloor \log \frac{e^{\psi(z^t, z^+)}}{\frac{1}{n} \sum_{i=1}^n e^{\psi(z^t, z_i^+)}} \right\rfloor
\end{aligned}
\tag{12}
$$

the second step is derived from the Jensen's inequality, where $p(z^+)$ approximates $\pi(z^+)$. We construct the similarity loss by maximizing mutual information, which is implemented by minimizing a negative lower bound of mutual information. The similarity loss can be expressed as:

$$\ell^{\text{sim}} = -I_{\text{Low}}(z^+; z^t) = -\frac{1}{n} \sum_{j=1}^n \log \frac{e^{\psi(z_j^t, z_j^+)}}{\sum_{i=1}^n e^{\psi(z_j^t, z_i^+)} + \xi} \tag{13}$$

**Self-supervised difference loss.** Similarly, we construct the difference loss by minimizing mutual information. In practice, researchers often use an upper bound of mutual information as an approximation for this minimization. By introducing a random variable $\mathcal{N}$ (negative samples representations in this paper) and applying the definition of mutual information, we obtain:

$$I(z^-; z^t) = I(z^-; z^t; \mathcal{N}) + I(z^-; z^t \mid \mathcal{N}) \tag{14}$$

and $I(z^-; z^t \mid \mathcal{N})$ is the conditional mutual information. Since $z^-$ is sampled from the set $\mathcal{N}$ and is conditionally independent of the current samples feature set $z^t$, we can deduce that:

$$
\begin{aligned}
I(z^-; z^t \mid \mathcal{N}) &= H(z^- \mid \mathcal{N}) - H(z^- \mid \mathcal{N}, z^t) \\
&= H(z^- \mid \mathcal{N}) - H(z^- \mid \mathcal{N}) = 0
\end{aligned}
\tag{15}
$$

$H(\cdot)$ denotes the information entropy. By combining Eq. 14 and Eq. 15, we can derive:

$$
\begin{aligned}
I(z^-; z^t) &= I(z^-; z^t; \mathcal{N}) \\
&= I(\mathcal{N}; z^-) - I(\mathcal{N}; z^- | z^t) \\
&= I(\mathcal{N}; z^-) + I(\mathcal{N}; z^t) - I(\mathcal{N}; z^-, z^t)
\end{aligned}
\tag{16}
$$

The third step is derived using the chain rule of mutual information. Based on the variational information bottleneck theory, the variational upper bounds of $I(\mathcal{N}; z^-)$ and $I(\mathcal{N}; z^t)$ can be obtained, yielding an upper bound of the mutual information $I(z^-; z^t)$ as follows:

$$
\begin{aligned}
I_{Up}(z^-; z^t) &= I(\mathcal{N}; z^-) + I(\mathcal{N}; z^t) - I(\mathcal{N}; z^-, z^t) \\
&\leq D_{KL}(p(z^- \mid \mathcal{N}) \,||\, q(z^-)) + D_{KL}(p(z^t \mid \mathcal{N}) \,||\, q(z^t)) \\
&\quad - \mathbb{E}_{q(z^- | \mathcal{N}) q(z^t | \mathcal{N})}[\log p(\mathcal{N} \mid z^-, z^t)]
\end{aligned}
\tag{17}
$$

The difference loss is approximated as:

$$
\ell^{diff} \approx D_{KL}(p(z^- \mid \mathcal{N}) \,||\, q(z^-)) + D_{KL}(p(z^T \mid \mathcal{N}) \,||\, q(z^t))
\tag{18}
$$

### A.2 PROOF OF THEOREM 1

**Lemma 1:** For any $\delta > 0$, assuming the model has previously converged on a stationary distribution $n > 10 d^{orth} \log \frac{2}{\delta}$, there exists at least a probability $1 - \delta$ such that the loss under selective fine-tuning becomes zero (Lee et al.):

$$
\mathcal{L}^{ft}(\theta_\infty^{sle}, \theta_\infty^{oth}; \mathcal{D}^t) = 0
\tag{19}
$$

**Proof:** When only the selected layers are updated, the remaining frozen layers remain unchanged, so we have $\theta_\infty^{oth} = \theta_0^{oth}$. These layers stay frozen during the fine-tuning process. The loss function of the model on the current samples $\mathcal{D}^t$ is defined as:

$$
\mathcal{L}^{ft}(\boldsymbol{\theta}_\infty^{sle}, \boldsymbol{\theta}_0^{oth}; \mathcal{D}^t) = \frac{1}{n} \sum_{i=1}^{n} \ell(f(x_i^t; \boldsymbol{\theta}_\infty^{sle}, \boldsymbol{\theta}_0^{oth}), y_i^t)
\tag{20}
$$

$f(\cdot; \theta)$ denotes the forward propagation function, and $\ell(\cdot)$ is the squared loss. We set the model's output layer as a linear layer and freeze the model parameters $\theta_0^{oth}$, so the loss function becomes a convex function with respect to $\theta_\infty^{sle}$. The model output is expressed as:

$$
f(x_i^t; \boldsymbol{\theta}_\infty^{sle}, \boldsymbol{\theta}_0^{oth}) = \boldsymbol{\theta}_\infty^{sle} \cdot \boldsymbol{\phi}(x_i^t; \boldsymbol{\theta}_0^{oth})
\tag{21}
$$

$\phi(x_i^t; \theta_0^{oth})$ is the nonlinear transformation from the frozen layers, then:

$$
\mathcal{L}^{ft}(\theta_\infty^{sle}) = \sum_{i=1}^{n} (\theta_\infty^{sle} \cdot \phi(x_i^t) - y_i^t)^2
\tag{22}
$$

The Eq. 22 is a convex function with respect to $\theta_\infty^{sle}$. This means the final parameters will converge to a global minimum, i.e., $\mathcal{L}^{ft}(\theta_\infty^{sle}, \theta^{oth}; \mathcal{D}^t) = 0$. Moreover, since $\theta_\infty^{oth} = \theta_0^{oth}$, we have $\mathcal{L}^{ft}(\theta_\infty^{sle}, \theta_0^{oth}; \mathcal{D}^t) = 0$.

**Lemma 2:** With at least probability 1, full fine-tuning yields a non-zero loss at all times.

$$
\mathcal{L}^{ful}(\boldsymbol{\theta}_\infty^{sle}, \boldsymbol{\theta}_\infty^{oth}; \mathcal{D}^t) > 0
\tag{23}
$$

**Proof:** Suppose the model function space is $\mathcal{F} = \{ f_\theta : \theta \in \Theta \}$, and the current samples $\mathcal{D}^t$ satisfy the mapping $y_i^t = f_*^t(x_i^t)$, with a probability distribution of $P_t(x, y)$. The mapping between features and labels in historical samples is given by $y_i^m = f_*^m(x_i^m)$, after concept drift occurs, we have $y_i^t = f_*^t(x_i^t) \neq y_i^m = f_*^m(x_i^m)$, due to the limited representation capacity of the model, which cannot adapt to the new data distribution in time. Therefore, $f_*^t(x_i^t) \notin \mathcal{F}$. The expected squared loss of the model under the new distribution is:

$$
\mathcal{L}^{ful}(\theta^{sle}, \theta^{oth}; \mathcal{D}^t) = \mathbb{E}_{x \sim P_t(x)}[(f_\theta(x) - f_*^t(x))^2]
\tag{24}
$$

Full fine-tuning means adjusting all parameters $\theta^{all} = \theta^{sle} + \theta^{oth}$ to minimize the loss:

$$\mathcal{L}_*^{ful}(\theta^{\text{all}}; \mathcal{D}^t) = \inf_{\theta \in \Theta} \mathcal{L}_*^{ful}(\theta^{\text{sle}}, \theta^{\text{oth}}; \mathcal{D}^t) \tag{25}$$

since $f_*^t(x_i^t) \notin \mathcal{F}$, the model incurs an approximation error:

$$\epsilon_{approx} := \inf_{\theta \in \Theta} \mathbb{E}_{x \sim P_t(x)}[(f_\theta(x) - f_*^t(x))^2] > 0 \tag{26}$$

Therefore $\mathcal{L}_\infty^{ft}(\theta^{\text{sle}}, \theta^{\text{oth}}; \mathcal{D}^t) > 0$, so Lemma 2 holds. Based on Lemma 1 and Lemma 2, we have: $\mathcal{L}^{ful}(\theta^{all}, \mathcal{D}^t) \geq \mathcal{L}^{ft}(\theta^{sle}, \mathcal{D}^t) = 0, \forall t$. Therefore, Theorem 1 holds.

## A.3 ANALYSIS OF THE REGRET BOUNDARY

**Proposition 1:** The loss function $f(\theta)$ of ATF-Net is strongly convex and satisfies the following inequality for any parameters $\theta_1, \theta_2$: $f(\theta_1) \geq f(\theta_2) + \nabla f(\theta_1)^T (\theta_2 - \theta_1)$.

***Proof:*** As can be seen from the last paragraph of Section 3.3, the loss function $f(\theta)$ can be expressed as: $f(\theta) = \mathcal{L}^{KD} + R(\varphi, \theta)$. The loss function consists of KL divergence and regularization terms, the regularization term is the $L2$ norm, and it is well known that the $L2$ norm is a strong convex function. When $\mathcal{L}^{KD}$ is a convex function, it can be proved that $f(\theta)$ is strongly convex. We use the $P$ and $Q$ to represent the probability distributions, from KL divergence:

$$\mathcal{L}^{KD} = D_{KL}(P\|Q) = \sum_{x^T} P(x^T) \log(\frac{P(x^T)}{Q(x^T)}) \tag{27}$$

$x^T$ represents current samples. Assuming $D_{KL}(P, Q)$ is a convex function, since KL divergence does not satisfy triangular symmetry, and we use MRN-Net to help fit the AFT-Net, so let $Q$ be a fixed term. From the properties of convex functions, we know:

$$D_{KL}((\lambda P_1 + (1 - \lambda)P_2) \| Q) \leq \lambda D_{KL}(P_1 \| Q) + (1 - \lambda)D_{KL}(P_2 \| Q) \tag{28}$$

where $\lambda \in [0, 1]$ is the weight factor and $P_1, P_2$ are arbitrary distributions. If Eq. 28 holds, it can be proved that $\mathcal{L}^{KD}$ is a convex function. Let $P_\lambda = \lambda P_1 + (1 - \lambda)P_2$, expand the left side of Eq. 28 to:

$$D_{KL}(P_\lambda \| Q) = \sum_{x^T} P_\lambda(x^T) \cdot \log \frac{P_\lambda(x^T)}{Q(x^T)} \tag{29}$$

For ease of calculation, we use $F(P_\lambda) = P_\lambda \cdot \log \frac{P_\lambda}{Q}$, $Q$ is a fixed term, so $F(P_\lambda)$ is a function about $P_\lambda$, its second derivative is:

$$F'(P_\lambda) = \log P_\lambda + 1 - \log Q, F''(P_\lambda) = \frac{1}{P_\lambda} \tag{30}$$

$\frac{1}{P_\lambda}$ is the probability distribution for $z^{AFT}$, so $1/P_\lambda > 0$, therefore $F''(P_\lambda) > 0$ and $F(P_\lambda)$ is a convex function. From Jensen's inequality we know:

$$\begin{aligned} F(P_\lambda(x^T)) &= P_\lambda(x^T) \cdot \log \frac{P_\lambda(x^T)}{Q(x^T)} \\ &\leq \lambda P_1(x^T) \log \frac{P_1(x^T)}{Q(x^T)} + (1 - \lambda)P_2(x^T) \log \frac{P_2(x^T)}{Q(x^T)} \end{aligned} \tag{31}$$

The sum of all samples is known:

$$\begin{aligned} \sum_{x^T} P_\lambda(x^T) \cdot \log \frac{P_\lambda(x^T)}{Q(x^T)} &\leq \lambda \sum_{x^T} P_1(x^T) \log \frac{P_1(x^T)}{Q(x^T)} \\ &\quad + (1 - \lambda) \sum_{x^T} P_2(x^T) \log \frac{P_2(x^T)}{Q(x^T)} \\ &= \lambda D_{KL}(P_1 \| Q) + (1 - \lambda)D_{KL}(P_2 \| Q) \end{aligned} \tag{32}$$

Eq. 28 holds, i.e. $\mathcal{L}^{KD} = D_{KL}(P||Q)$ is a convex function of $P$. And because $R(\varphi, \theta)$ is a strong convex function, so the loss $f(\theta)$ of the AFT-Net is a strong convex function. It satisfies all properties of strong convex functions and provides a guarantee for the proof of sublinear regret bounds.

## A.4 PROOF OF REGRET BOUNDARY

From **Assumption 1**, we know that the gradient of the AFT-Net is bounded, i.e. $g_t = ||\nabla f(\theta)|| \leq l$. And according to **Assumption 2**, the diameter of the parameter domain is $\Gamma$, so the gradient boundary of $R(\varphi, \theta)$ is:

$$|| \nabla R(\varphi, \theta) || = \frac{\beta_1}{2} \nabla || \varphi - \theta ||^2 \leq \beta_1 \Gamma \tag{33}$$

Then $g_t = l = l_1 + \beta_1 \Gamma$, $l_1$ is the boundary of $||\nabla \mathcal{L}^{KD}||$. Eq. 9 can be transformed into:

$$regret = \sum_{t=1}^{T} f_t(\theta_t) - \min_{\theta \in \mathcal{W}} \sum_{t=1}^{T} f_t(\theta) = \sum_{t=1}^{T} f_t(\theta_t) - \sum_{t=1}^{T} f_t(\theta_*)$$

$$= \sum_{t=1}^{T} (f_t(\theta_t) - f_t(\theta_*)) \tag{34}$$

According to (Cesa-Bianchi & Lugosi, 2006), we set the learning rate to $\eta_t = 1/(\delta t)$, from the previous section, we can see that $f(\theta)$ is a strong convex function, according to its nature, it can be obtained:

$$f_t(\theta_t) - f_t(\theta_*) \leq \langle \nabla f_t(\theta_t), \theta_t - \theta_* \rangle - \frac{\delta}{2} ||\theta_t - \theta_*||^2$$

$$\leq \frac{1}{2\eta_t} (||\theta_t - \theta_*||^2 - ||\theta_{t+1} - \theta_*||^2 \tag{35}$$

$$+ \frac{\eta_t}{2} (l_1 + \beta_1 \Gamma)^2 - \frac{\delta}{2} || \theta_t - \theta_* ||^2)$$

When we sum them over the $T$-round iteration, we get:

$$\sum_{t=1}^{T} (f_t(\boldsymbol{\theta}_t) - f_t(\boldsymbol{\theta}_*)) \leq \frac{1}{2\eta_1} ||\theta_1 - \theta_*||^2 - \frac{\delta}{2} ||\theta_1 - \theta_*||^2$$

$$- \frac{1}{2\eta_T} || \theta_{T+1} - \theta_* ||^2$$

$$+ \frac{1}{2} \sum_{t=2}^{T} \left( \frac{1}{\eta_t} - \frac{1}{\eta_{t-1}} - \delta \right) ||\theta_t - \theta_*||^2 \tag{36}$$

$$+ \frac{(l_1 + \boldsymbol{\beta_1}\Gamma)^2}{2} \sum_{t=1}^{T} \boldsymbol{\eta}_t$$

Substituting $\eta_t$ into Eq. 36 yields:

$$\sum_{t=1}^{T} \left( f_t(\theta_t, D^T) - f_t(\theta_*, D^T) \right) \leq \frac{(l_1 + \beta_1 \Gamma)^2}{2\delta} \sum_{t=1}^{T} \frac{1}{t}$$

$$\leq \frac{(l_1 + \beta_1 \Gamma)^2}{2\delta} (\ln T + 1) \tag{37}$$

Thus, the regret boundary can be expressed as:

$$regret = \sum_{t=1}^{T} f_t(\theta_t) - \min_{\theta \in \mathcal{W}} \sum_{t=1}^{T} f_t(\theta)$$

$$\leq \frac{(l_1 + \beta_1 \Gamma)^2}{2\delta} (\ln T + 1) \approx O(\frac{(l_1 + \beta_1 \Gamma)^2}{2\delta} \ln T) \tag{38}$$

$\beta_1$ is the weight factor of the regularization penalty term in the loss function, and $\delta$ is the initial learning rate adjustment factor, which decreases with time. $regret/T$ is 0 as $T$ approaches infinity, meaning that our model converges within $T$ steps.

# B    ADDITIONAL EXPERIMENTAL RESULTS

## B.1    EXPERIMENTAL SETTINGS

MetaOCDN is implemented using the deep learning framework PyTorch. The experimental environment is as follows: Intel(R) Xeon(R) Platinum 8468V, 1.0TB memory and NVIDIA H100 graphics card. Furthermore, all of our experiments follow the standard setting of stream data prequential (Brzezinski & Stefanowski, 2014), that is, the data of each batch is first used to test the model and then to train the model, and each dataset passes through the model only once.

In this paper, ResNet with 12 layers is adopted as baseline, dense blocks are constructed by using two-layer one-dimensional convolution Conv1d and ReLU, and channel attention and spatial attention modules are added after each dense block to improve the perception ability of the model for key information. In addition, considering the limitation of memory resources, we set the size of historical samples to $m = 20$, which means that the samples of the last 20 batches are stored in the memory module, the constant offset term of similarity loss $\xi$ is set to 0.001, and the initial value of the weight factor of regularization penalty term $\beta_1$ is 1e-4.

## B.2    DATASETS

In order to verify the performance of MetaOCDN under different tasks, we investigated the classical datasets of concept drift in classification task and regression task, respectively.

**Classification Datasets:** We used the data flow generator in the Massive Online Analysis (MOA) platform (Bifet & Gavalda, 2007) to generate three abrupt, gradual, and incremental concept drift datasets: *RBFBlips*, *Sea* and *Hyperplane*. For convenience of testing, we set the drift sites as 25K, 50K and 75K. Furthermore, we also selected three real datasets: *Kddcup99*, *MIRS* (Krüger et al., 2016) and *Yoga* (Krüger et al., 2016).

**Regression Datasets:** For the regression task, we tested MetaOCDN and other methds on a series of time series prediction datasets: *ETTH2*, *ETTm1* and *WTH* (Zhou et al., 2023). These datasets are real datasets, and the details of the datasets are shown in Table B.2.

Table 3: Characteristics of Datasets

|  | Datasets | Instances | Features | Target variable | Types | Number Of drift |
|---|---|---|---|---|---|---|
| **Class.** | *RBFblips* | 100K | 20 | 4 | Abrupt | 3 |
|  | *Sea* | 100K | 3 | 2 | Gradual | 3 |
|  | *Hyperplane* | 100K | 10 | 2 | Incremental | - |
|  | *Kddcup99* | 4.94M | 23 | 23 | Unknown | - |
|  | *MIRS* | 4260 | 3600 | 2 | Abrupt | - |
|  | *Yoga* | 3300 | 426 | 2 | Unknown | - |
| **Reg.** | *ETTH2* | 17420 | 6 | 1 | Unknown | - |
|  | *ETTm1* | 69680 | 6 | 1 | Unknown | - |
|  | *WTH* | 35065 | 11 | 1 | Unknown | - |

## B.3    COMPARISON METHODS

Furthermore, we compare OCF with various methods, including traditional concept drift adaptive method: DWM (Kolter & Maloof, 2007): Dynamic Weighted Majority (DWM) is an ensemble method for handling concept drift. It continuously trains online learners, dynamically adjusts their weights based on performance. OBC (Oza & Russell, 2001): Bagging and boosting are ensemble methods that combine multiple base learners to improve performance. RUS (Wang & Pineau, 2016): RUS combines online ensemble techniques with cost-sensitive strategies from batch learning, resulting in theoretically sound algorithms with guaranteed convergence under certain conditions. LEV (Bifet et al., 2010): LEV adapts classical ensemble methods like bagging, boosting, and Random Forests to evolving data streams by introducing additional randomization to inputs and outputs while preserving bagging's simplicity. ARF (Gomes et al., 2017): Adaptive Random Forest (ARF) extends Random Forests to data streams by introducing adaptive mechanisms and resampling strategies to handle concept drift effectively.

And some deep neural networks: DNN (Guo et al., 2016): The DNN is the most common network. ResNet (He et al., 2016): ResNet alleviates the vanishing gradient problem in deep networks by introducing skip connections and allowing cross-layer information transmission. Highway (Srivastava et al., 2015): Highway networks introduce adaptive gating units to regulate information flow across many layers, enabling the direct training of extremely deep networks using simple gradient descent. HBP (Sahoo et al., 2017): Hedge Backpropagation (HBP) for effectively updating DNN parameters in online learning settings. DenseNet (Huang et al., 2019): DenseNet promotes feature reuse and alleviates the vanishing gradient problem by connecting the outputs of each layer with those of all the previous layers.

We have also introduced the latest time series prediction methods: Informer (Zhou et al., 2021): Informer is an efficient Transformer model. By introducing the ProbSparse self-attention mechanism, self-attention distillation and generative decoder, it solves the computational and structural bottleneck problems of Transformer in long sequence time series prediction. ER (Chaudhry et al., 2019): ER stores the previous data in the buffer and interweaves it with newer samples during the learning period. DER++ (Buzzega et al., 2020): DER++ adds the knowledge distillation strategy on the basis of ER. FsNet (Pham et al., 2022): FSNet is an online time series prediction framework inspired by the complementary learning system theory. By introducing layer-by-layer adaptors and associative memory mechanisms. Time-TCN (Bai et al., 2018): Time-TCN is a convolutional neural network structure in the time dimension. PatchTST (NIEY et al., 2023): PatchTST is an efficient modeling method for Transformer time series. It is independently designed by using time series slices as input tokens and channels to improve the prediction of long sequences and the learning effect of self-supervised representations, while reducing the computational cost of attention.

In all of these methods, the batch size is uniformly set to 100 and the hidden node is 100, using the ReLU activation function and a fixed learning rate of 0.01.

### B.4 EVALUATION INDICATORS

To measure OCF performance on different datasets, we use Average Real Accuracy (*Avgracc*) and Final Cumulative Accuracy (*Fincacc*) on the categorical datasets, and Mean Square Error (MSE) and Mean Absolute Error (MAE) on the regression datasets, respectively. And on all types of datasets, we adopted the Bonferroni-Dunn test to compare the differences among different methods. On the dataset with known drift sites, we used Recovery speed under accuracy (*RSA*) to test the convergence performance of different methods. Since MSE and MAE adopt common settings, we will not introduce them here. The specific details of the evaluation indicators are as follows:

(1) Average real accuracy (*Avgracc*): The average of the real-time accuracy of the model at each time step, which reflects the real-time performance of the model:

$$Avgracc = \frac{1}{T} \sum_{t=1}^{T} acc_t \tag{39}$$

where $acc_t$ is the real-time accuracy of the $t$-step time. The real-time accuracy of the model in this paper is adopted Class Balance Accuracy (CBA).

$$acc = CBA = \frac{\Sigma_{i=1}^{k} \frac{c_{ii}}{max(c_{i*}, c_{*i})}}{k} \tag{40}$$

where $k$ is the total number of categories, $c_{ii}$ is the $i$th element on the main diagonal of the prediction result confusion matrix, $c_{i*}$ and $c_{*i}$ represent one element in row $i$ and column $i$. The performance metric bias caused by class imbalance is mitigated by calculating class balance accuracy.

(2) Final cumulative accuracy (*Fincacc*): The ratio of the number of samples cumulatively predicted correctly to the number of samples cumulatively acquired up to the current time, which reflects the population of the model performance:

$$Fincacc = \frac{1}{T * n} \sum_{t=1}^{T} n_t \tag{41}$$

where $n$ represents the size of samples obtained at each timestamp, $n_t$ represents the number of samples for which the classifier predicts the correct label at the $t$th timestamp.

(3) Recovery speed under accuracy (*RSA*): An online learning model with good convergence can not only converge to the stable state of the new distribution in a short time after concept drift but also maintain the minimum real-time error during the convergence process. Therefore, the *RSA* is defined in the following way to measure the convergence performance of the model:

$$RSA = step * \epsilon_{avg} \tag{42}$$

where the $step$ denotes the number of time steps required from the concept drift site to the convergence site, and $\epsilon_{avg}$ denotes the average real-time error rate of the convergence process. For the definition of a convergence site, on the one hand, the amplitude of data fluctuation should not be too large, and at the same time, the randomness of data fluctuation should be considered. Therefore, this paper adopts the testing results of 20 subsequent reference sites of a certain site to define whether the site is a convergence site. If the accuracy difference between this site and subsequent reference sites is less than the given threshold, and the average accuracy of the first and last 10 reference sites of the reference sites is also less than the threshold, then the site is considered the convergence site. $\forall i, i \in \{1, \cdots, 20\}$,

$$|acc_t - acc_{t+i}| < \varepsilon \text{ and } \left| \frac{1}{10} \sum_{j=1}^{10} acc_{t+j} - \frac{1}{10} \sum_{k=1}^{20} acc_{t+k} \right| < \varepsilon \tag{43}$$

here, $\varepsilon$ is the convergence threshold parameter.

(4) In addition, the critical difference (CD) of all methods was calculated by the Bonferroni-Dunn test [31] to show the relative performance between the proposed and the comparison method. The performance of two classifiers is significantly different if the corresponding average rank sum differs by at least the critical difference:

$$CD = q_\alpha \sqrt{\frac{k(k+1)}{6N}} \tag{44}$$

where $q_\alpha$ is the critical value at significance level $\alpha$.

### B.5 ANALYSIS OF EXPERIMENTAL RESULTS

**Results on classification datasets.** Fig. 7 presents the *Fincacc* results of all methods on real-world datasets, showing that MetaOCDN achieves superior predictive accuracy.

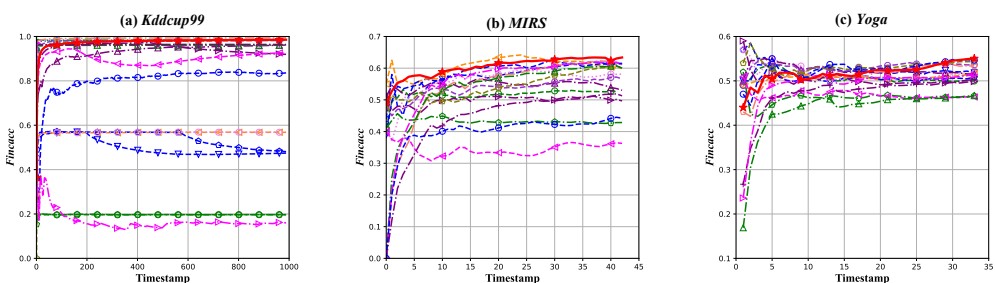

Figure 7: Comparison of *Fincacc* of different methods on real-world datasets

**Results on regression datasets.** Since traditional classification methods perform poorly on time series regression datasets, we only compare the methods with relatively better performance. The *MAE* results are shown in Fig. 8. As illustrated in the figure, MetaOCDN achieves strong performance across all three datasets. This is because MetaOCDN is capable of learning structured knowledge from historical samples. Time series data often contain global patterns within historical observations, and MetaOCDN leverages the MRN-Net to effectively capture long-term dependencies in the data, leading to superior results.

### B.6 SUPPLEMENTARY RESULTS OF THE ABLATION STUDY

**Gradient-aware Selective Fine-tuning analysis.** From Fig. 9, we observe that on the *Sea* dataset, when concept drift occurs, the gradient norms of Residual Block 1 and Residual Block 2 fluctu-

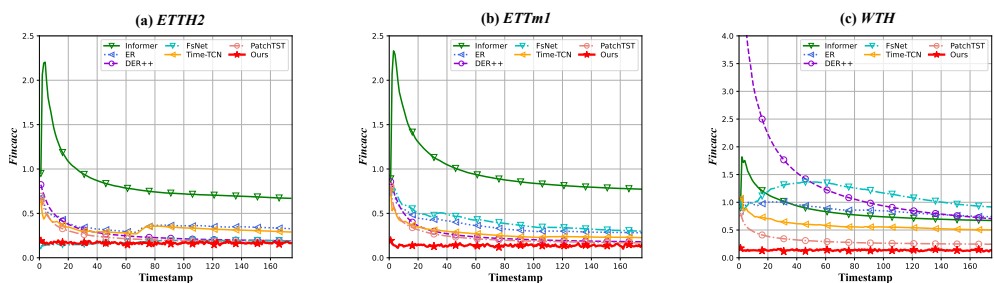

Figure 8: Comparison of *MAE* of different methods on real-world datasets

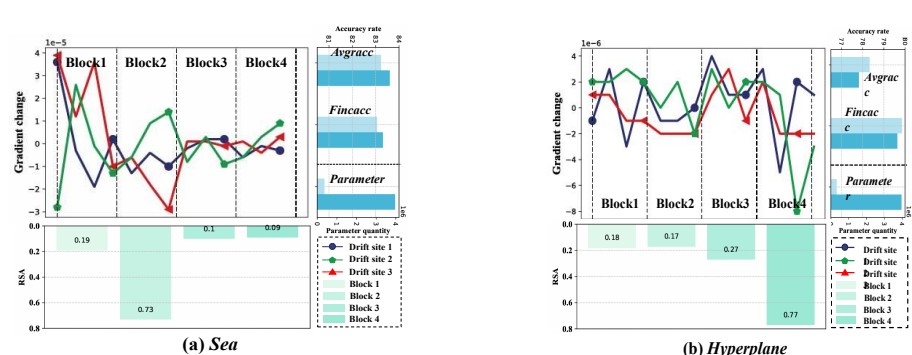

Figure 9: Gradient dynamics and result analysis

ate most significantly, while those of Residual Block 3 and Residual Block 4 remain nearly zero, showing almost no impact. This indicates that the first two residual blocks are more sensitive to distributional shifts and primarily contribute to adapting and representing drift patterns. Furthermore, combining this with the convergence speed results (bottom-left subfigure), we find that freezing Residual Blocks 1 and 2 leads to a significant decline in convergence speed, with the effect being particularly pronounced when Residual Block 2 is frozen; in contrast, freezing Residual Block 3 has almost no negative impact on convergence. This phenomenon further validates the critical role of Residual Blocks 1 and 2 in adapting to concept drift. On the other hand, the top-right subfigure shows that under the selective fine-tuning strategy, the model achieves accuracy performance (in terms of both average real-time accuracy and cumulative accuracy) comparable to full fine-tuning, while significantly reducing parameter overhead. This demonstrates that the strategy achieves a better trade-off between accuracy and efficiency, thereby enhancing resource utilization and deployment flexibility.

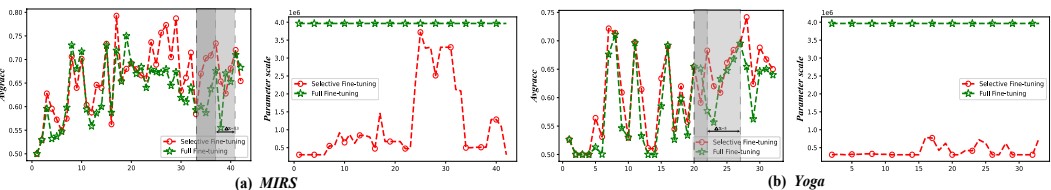

Figure 10: Convergence Speed and Parameter Scale Comparison

Fig. 10 illustrates the convergence speed and parameter scale of MetaOCDN on the *MIRS* and *Yoga* datasets. The experimental results show that the selective fine-tuning strategy helps MetaOCDN achieve faster convergence while requiring fewer parameters, thereby reducing computational overhead to some extent.

**Robustness Analysis of MRN-Net.** Specifically, we selected three datasets with clearly defined drift points and compared the performance of MetaOCDN with and without MRN-Net assistance after concept drift occurred. The evaluation metrics include *RSA* (Recovery Speed after Adaptation), which measures the model's real-time convergence ability during drift, and *DCE* (Drift Cumulative Error), which quantifies the accumulated error during the drift adaptation phase. The experimental results are shown in Fig. 11.

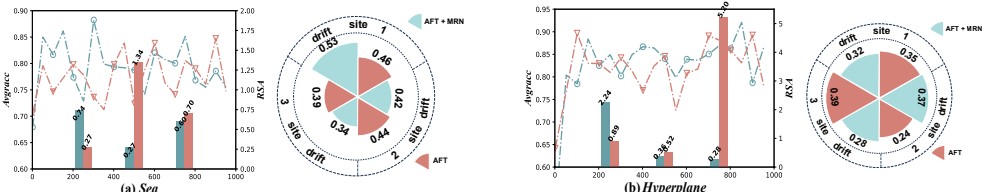

Figure 11: Comparison of *MAE* of different methods on real-world datasets

As shown in the figures, on the *RBFBlips*, *Sea*, and *Hyperplane* datasets with known drift points, MetaOCDN with the collaboration of AFT-Net and MRN-Net exhibits significantly smaller overall accuracy fluctuations compared to MetaOCDN relying solely on AFT-Net. When concept drift occurs, MRN-Net provides more stable initialization or adjustment signals for the online adaptation process, enabling the model to converge more rapidly to the new data distribution while substantially reducing error accumulation during the drift adaptation phase. Furthermore, this mechanism not only enhances the model's dynamic responsiveness and error suppression ability but also demonstrates consistent and notable advantages in stability and adaptability across multiple non-stationary environments, thereby validating the critical role of the MRN-Net in strengthening model robustness.

## B.7 THE USE OF LLMS

No large language models were used in the experiments or in writing this paper.

