# OpenReview forum: "MetaOCDN: A Cognition-Inspired Meta Optimized Complementary Dual Networks for Online Continual Concept Drift Adaptation"
_ICLR.cc/2026/Conference — ICLR 2026 Conference Withdrawn Submission_

### Official Review · Reviewer_WAmY · 2025-10-31

**Soundness:** 3
**Presentation:** 2
**Contribution:** 3
**Rating:** 6
**Confidence:** 2

**Summary:**

This paper addresses the challenge of **concept drift** in streaming data mining. Inspired by the **Complementary Learning Systems (CLS)** theory in the human brain, the authors propose a dual-network architecture called **MetaOCDN**. The architecture consists of two components:

1. **Adaptive Fine-Tuning Network (AFT-Net):** Modeled after the hippocampus, it quickly adapts to new data distributions using a **gradient-aware selective fine-tuning** strategy.
2. **Meta-Representation Network (MRN-Net):** Modeled after the neocortex, it extracts stable, structured knowledge from historical data via **self-supervised learning**.

The authors claim that these two networks collaborate through **MAML-based knowledge distillation**, achieving **rapid adaptation (plasticity)** while maintaining **generalization (stability)**, and outperforming existing methods across multiple benchmarks.

**Strengths:**

1. **Strong empirical results:** The paper's most prominent strength is its significant performance improvement over **17 baseline methods** across **9 diverse datasets** (including classification and regression), as shown in Table 1. Its **average rank (Avg Rank)** is 2.55, demonstrating a clear advantage.

2. **Excellent ablation studies:** Section 5.2 is particularly well-executed. Through ablation experiments, the authors clearly show that AFT-Net's **gradient-aware fine-tuning** (Figures 5 and 6a) and MRN-Net's **stable anchor mechanism** (Figure 6b) are both critical to its high performance.

3. **Theoretical support:** The paper provides theoretical analysis in Section 4, including **Theorem 1** (demonstrating the superiority of selective fine-tuning) and **sublinear regret bounds** (proving convergence efficiency).

4. **Candid discussion of limitations:** In Section 5.1, the authors openly discuss the model’s failures on the **Hyperplane** (incremental drift) and **Kddcup99** (discrete features) datasets, providing logical explanations for these shortcomings.

**Weaknesses:**

1. **Presentation is confusing:** As noted under "Presentation Quality," the authors’ heavy reliance on CLS theory (as a narrative device) **obfuscates a clear understanding of the technical contribution**.

2. **Limited conceptual novelty:** The paper’s main weakness lies in its core inspiration. The CLS-inspired dual-network (fast/slow) architecture has been explored in the **continual learning literature** (e.g., DualNet). The authors should clarify in the related work section how their approach **differs from these prior CLS-inspired models**.

3. **Limitations of the strategy:** As acknowledged by the authors, AFT-Net's **gradient-aware** strategy fails in cases of very slow or subtle **incremental drift** (e.g., the Hyperplane dataset). This suggests that the strategy **relies on sufficiently strong gradient signals**, which limits its general applicability.

**Questions:**

1. The core idea of a CLS-inspired fast/slow dual-network already exists in prior work (e.g., DualNet). Could the authors more clearly explain how your contribution—such as the **specific implementations of AFT-Net and MRN-Net**—differs from these existing CLS-inspired architectures, and what the **key distinctions and advantages** are?

2. Given that the **gradient-aware** strategy fails under subtle incremental drift (e.g., Hyperplane), does this imply a fundamental limitation? Have you considered potential solutions, perhaps by introducing a mechanism that does **not entirely rely on strong gradient signals** to complement AFT-Net’s activation?

---

### Official Review · Reviewer_dxkh · 2025-10-31

**Soundness:** 2
**Presentation:** 2
**Contribution:** 2
**Rating:** 2
**Confidence:** 4

**Summary:**

This paper proposes MetaOCDN, a novel approach for online concept drift adaptation inspired by the Complementary Learning Systems (CLS) theory from neuroscience. The authors draw parallels between the human brain's hippocampus-neocortex collaboration and a dual-network architecture for streaming data. Specifically, MetaOCDN consists of two components: (1) an Adaptive Fine Tuning Network (AFT-Net) that simulates the hippocampus for rapid adaptation to new concepts through gradient-aware selective fine-tuning, and (2) a Meta Representation Network (MRN-Net) that simulates the neocortex for extracting structured knowledge using self-supervised duality loss. The authors further design a MAML-based multi-scale knowledge distillation strategy to facilitate knowledge transfer between these networks. The paper includes theoretical analysis showing sublinear regret bounds and extensive experimental validation across multiple classification and regression tasks with concept drift, demonstrating consistent improvements over state-of-the-art baselines.

**Strengths:**

1. The paper makes a compelling connection between neuroscience (CLS theory) and machine learning for concept drift adaptation, providing a biologically inspired framework that addresses a critical challenge in streaming data analysis.

2. The gradient-aware selective fine-tuning strategy represents a significant contribution. The authors' insight that different layers respond differently to concept drift (demonstrated in Figure 2) leads to a more efficient adaptation mechanism than full fine-tuning.

**Weaknesses:**

1. While the paper acknowledges poor performance on the Hyperplane dataset (incremental drift), it lacks sufficient analysis of why this occurs and how the method could be adapted for this important drift type. The explanation that "the AFT-Net tends to freeze more layers" is insufficient without concrete evidence or proposed solutions.

2. The paper doesn't provide a detailed analysis of the computational and memory overhead introduced by maintaining two networks and the knowledge distillation process. This is crucial for real-world deployment considerations.

3. There's limited discussion about the sensitivity of the method to key hyperparameters (e.g., β for balancing similarity/difference loss, the drift-aware threshold parameters). Understanding how these parameters affect performance would strengthen practical applicability.

4. While the paper compares with many baselines, it's unclear if the most recent state-of-the-art methods specifically designed for concept drift (from the last 6-12 months) are included in the comparison.

5. The paper lacks discussion about practical constraints such as memory requirements for storing historical samples, computational constraints in real-time applications, and how the approach scales to high-dimensional data.

**Questions:**

- Theorem 1's claim that "full fine-tuning yields a non-zero loss at all times" seems overly strong. While the proof shows full fine-tuning may not reach zero loss due to approximation error, there are scenarios where full fine-tuning could converge to zero loss on the current distribution (though it would likely forget previous knowledge). This distinction should be clarified.

- There's insufficient analysis of failure cases beyond the incremental drift issue.

- More visualization of how the gradient-aware selective fine-tuning actually operates on different drift types would strengthen the claims.

- Technical sections are not easy to follow. It is suggested to improve the writing logic and add some figures to explain the method more clearly.

- Many grammatical issues exist (e.g., "we analysis" instead of "we analyze" on page 2).

- The connection between neuroscience concepts and technical implementation could be strengthened in the methodology section.

- The related work section can be improved from the following aspects. Some recent papers on concept drift from 2024 (beyond those cited) might be missing. The connection to continual learning literature could be expanded, as concept drift shares significant overlap with continual learning challenges. The paper could better position itself relative to meta-learning approaches specifically designed for concept drift adaptation.

---

### Official Review · Reviewer_djQt · 2025-11-01

**Soundness:** 3
**Presentation:** 3
**Contribution:** 4
**Rating:** 8
**Confidence:** 1

**Summary:**

This paper proposes a cognition-inspired meta-learning framework called MetaOCDN to address the problem of online continual learning under concept drift in open environments. The method draws on the human brain's Complementary Learning Systems (CLS) theory, analogizing the collaborative mechanism between the hippocampus and neocortex as two sub-networks: AFT-Net, which is responsible for rapidly adapting to new concept changes, and MRN-Net, which extracts stable structured knowledge from historical samples. These two networks interact through MAML-based multi-scale knowledge distillation. Extensive experiments show that MetaOCDN outperforms existing mainstream methods in both classification and regression tasks.

**Strengths:**

1.	The paper is highly innovative, combining the CLS theory from cognitive science to design the AFT-Net and MRN-Net dual-network architecture, and proposing the gradient-aware selective fine-tuning strategy.

2.	The theoretical analysis of the paper is rigorous: it provides proofs for the convergence and regret bound of the selective fine-tuning strategy, enhancing the theoretical depth.

3.	The paper validates the effectiveness of the MetaOCDN model on multiple standard datasets with different types of drift, and designs ablation experiments and convergence analysis, making the results convincing.

**Weaknesses:**

1.A brief explanation of the P_t (X，y) function in Section 3 is recommended. Additionally, the formulas in Sections 3.2-4.2 are dense, so it is suggested to include some intuitive illustrations to aid understanding.

2.Some details in the paper need attention. In Table 1, the ETtml column should indicate the second-best results, consistent with the other columns.

3.The paper proposes using the gradient-aware selective fine-tuning strategy to improve computational efficiency, but the computational complexity of MetaOCDN is not analyzed in detail in the paper.

**Questions:**

1. Learning the data stream should concern both the trade-off of accuracy and computational complexity. So, the computational complexity of the proposed learning method is required.

2. There are hyperparameters in the paper, like $\beta$, the author should explain how to establish the hyperparameter, and a parameter sensitivity analysis is needed.

3. Since the proposed method is designed based on MAML, please explain the difference between the proposed method and traditional multi-task learning method.

---

### Official Review · Reviewer_B8o5 · 2025-11-01

**Soundness:** 2
**Presentation:** 3
**Contribution:** 2
**Rating:** 4
**Confidence:** 4

**Summary:**

MetaOCDN is a framework for online learning under concept drift, where data distributions change over time. Authors draw inspiration from the human brain which learns using fast adaptation via the hippocampus and stable representations via the neocortex. The authors propose two specialized networks AFTnet and MRNnet. AFTnet adapts to new data using a gradiant aware selective fine tuning method that updates key layers affected by the drift. MRNnet learns generalizable representations using historical samples using a self supervised duality loss to improve stability and feature extraction over time. Both the networks interact through a MAML based knowledge distillation framework

**Strengths:**

- the motivation for the method is well explained, with the 2 networks mimicking hippocampus and neocortex functionality from brain
- comprehensive evaluation across both classification and regression drift benchmarks with ablation studies

**Weaknesses:**

-
- The paper does not benchmark against recent large-scale online continual or meta-learning approaches [1]. Also does not compare against CLS theory based continual learning frameworks [2]
- Most of the methods in comparison employ one network, so how to fairly compare against this method that utilized 2 networks and distillation approach?
- The MAML-based bi-level optimization and multi-scale distillation can be computationally heavy for high-dimensional streams;?
- Need more details about the replay buffer. How many samples, how does it affect the learning?
- Results depend heavily on the gradient sensitivity threshold and layer-freezing heuristic. Need more details on drift-aware threshold and its sensitivity.


[1] When Meta-Learning Meets Online and Continual Learning: A Survey
[2] Learning Fast, Learning Slow: A General Continual Learning Method based on Complementary Learning System

**Questions:**

- The authors argue that the full online gradient descent is O(d) complexity and that freezing layers improves response efficiency, but they do not provide analysis of computing per-layer gradients every step, maintaining the history matrix, and running the selection rule
- Lemma 1 uses d_orth but it is not defined anywhere.
- Line 242 - what are these multi-scale units and why we split it like that?
-  Why is sparsity done layer-wise and not at neuron-level?

---

### Note · Authors · 2025-12-04

I have read and agree with the venue's withdrawal policy on behalf of myself and my co-authors.